# Demystifying Adversarial Training via A Unified Probabilistic Framework

Yifei Wang [1]   Yisen Wang [2]   Jiansheng Yang [1]   Zhouchen Lin [2]

## Abstract

Adversarial Training (AT) is known as an effective approach to enhance the robustness of deep neural networks. Recently researchers notice that robust models with AT have good generative ability and can synthesize realistic images, while the reason behind it is yet under-explored. In this paper, we demystify this phenomenon by developing a unified probabilistic framework, called Contrastive Energy-based Models (CEM). On the one hand, we provide the first probabilistic characterization of AT through a unified understanding of robustness and generative ability. On the other hand, our CEM can also naturally generalize AT to the unsupervised scenario and develop principled unsupervised AT methods. Based on these, we propose principled adversarial sampling algorithms in both supervised and unsupervised scenarios. Experiments show our sampling algorithms significantly improve the sampling quality and achieves an Inception score of 9.61 on CIFAR-10, which is superior to previous energy-based models and comparable to state-of-the-art generative models.

## 1. Introduction

Adversarial Training (AT) is one of the most effective approaches developed so far to improve the robustness of deep neural networks (DNNs) (Madry et al., 2018; Wang et al., 2019). AT solves a minimax optimization problem, with the *inner maximization* generating adversarial examples by maximizing the classification loss, and the *outer minimization* finding model parameters by minimizing the loss on adversarial examples generated from the inner maximization. Recently, researchers have noticed that such robust classifiers obtained by AT are able to extract features that are perceptually aligned with humans (Engstrom et al., 2019). Furthermore, they are able to synthesize realistic images on

[1]School of Mathematical Sciences, Peking University, Beijing, China [2]Key Lab. of Machine Perception (MoE), School of EECS, Peking University, Beijing, China. Correspondence to: Yisen Wang <yisen.wang@pku.edu.cn>.

*Accepted by the ICML 2021 workshop on A Blessing in Disguise: The Prospects and Perils of Adversarial Machine Learning.* Copyright 2021 by the author(s).

par with state-of-the-art generative models (Santurkar et al., 2019). Nevertheless, it is still a mystery why AT is able to learn more semantically meaningful features and turn classifiers into generators. Besides, as generative methods, AT requires labeled data $\{(\mathbf{x}_i, \mathbf{y}_i)\}$ while canonical generative models only require unlabeled data, *e.g.*, VAE (Kingma & Welling, 2014) and GAN (Goodfellow et al., 2015). Thus, it is worth exploring if it is possible to train a robust model without labeled data. Several recent works (Jiang et al., 2020; Kim et al., 2020; Ho & Nvasconcelos, 2020) have proposed unsupervised AT by adversarially attacking the InfoNCE loss (Oord et al., 2018) (a widely used objective in unsupervised contrastive learning), which indeed improves the robustness of contrastive encoders. However, a depth investigation and understanding for unsupervised AT is still missing.

In this paper, we address the above issues by proposing a unified framework with the following contributions:

- **A unified probabilistic framework.** We propose Contrastive Energy-based Model (CEM) that incorporates both supervised and unsupervised learning paradigms. Our CEM provides a unified probabilistic understanding of previous standard and adversarial training methods in both supervised and unsupervised learning.

- **Demystifying AT.** Based on CEM, we propose the first probabilistic interpretation for AT, that is, it is inherently a (biased) maximum likelihood training of the corresponding energy-based model, which explains the generative ability of robust models learned by AT.

- **Principled unsupervised AT.** Specifically, under our proposed CEM framework, we establish the equivalence between the importance sampling of CEM and the InfoNCE loss of contrastive learning, which enables us to design principled adversarial training and sampling for unsupervised learning.

- **State-of-the-art adversarial sampling algorithms.** Inspired by this, we propose some novel sampling algorithms with better sample quality than previous methods. Notably, we show that our sampling methods achieve state-of-the-art sample quality (9.61 Inception score) with unsupervised robust models, which is comparable to both the supervised counterparts and other state-of-the-art generative models.

## 2. A Unified Probabilistic Framework

Our proposed CEM is a special kind of Enery-Based Models (EBMs) (LeCun et al., 2006) that models the joint distribution $p_{\boldsymbol{\theta}}(\mathbf{u}, \mathbf{v})$ with a contrastive similarity function $f_{\boldsymbol{\theta}}(\mathbf{u}, \mathbf{v})$,

$$p_{\boldsymbol{\theta}}(\mathbf{u}, \mathbf{v}) = \frac{\exp(f_{\boldsymbol{\theta}}(\mathbf{u}, \mathbf{v}))}{Z(\boldsymbol{\theta})}, \qquad (1)$$

where $\mathbf{u}, \mathbf{v}$ are two random variables, $\boldsymbol{\theta}$ are model parameters and $Z(\boldsymbol{\theta}) = \int \exp(f_{\boldsymbol{\theta}}(\mathbf{u}, \mathbf{v})) \, d\mathbf{u} d\mathbf{v}$ is the corresponding partition function. In CEM, the log probability $\log p_{\boldsymbol{\theta}}(\mathbf{u}, \mathbf{v})$ is proportional to the similarity between $\mathbf{u}$ and $\mathbf{v}$, as measured by the similarity function $f_{\boldsymbol{\theta}}(\mathbf{u}, \mathbf{v})$.

**Parametric CEM.** In the supervised scenario, we specify the Parametric CEM (P-CEM) that models the joint distribution $p_{\boldsymbol{\theta}}(\mathbf{x}, y)$ between data $\mathbf{x} \in \mathbb{R}^n$ and label $y \in [K]$,

$$p_{\boldsymbol{\theta}}(\mathbf{x}, y) = \frac{\exp(f_{\boldsymbol{\theta}}(\mathbf{x}, y))}{Z(\boldsymbol{\theta})} = \frac{\exp(g_{\theta}(\mathbf{x})^{\top} \mathbf{w}_y)}{Z(\boldsymbol{\theta})}, \qquad (2)$$

where $g_{\boldsymbol{\theta}} : \mathbb{R}^n \to \mathbb{R}^m$ denotes the encoder, $g(\mathbf{x}) \in \mathbb{R}^m$ is the representation of $\mathbf{x}$, and $\mathbf{w}_k \in \mathbb{R}^m$ refers to the parametric cluster center of the $k$-th class.

**Non-Parametric CEM.** In the unsupervised scenario, we do not have access to labels, thus we instead model the joint distribution between data $\mathbf{x}$ and representation $\mathbf{z}$ as

$$p_{\boldsymbol{\theta}}(\mathbf{x}, \mathbf{z}) = \frac{\exp(f_{\boldsymbol{\theta}}(\mathbf{x}, \mathbf{z}))}{Z(\boldsymbol{\theta})} = \frac{\exp(g_{\theta}(\mathbf{x})^{\top} \mathbf{z})}{Z(\boldsymbol{\theta})}, \qquad (3)$$

and the corresponding likelihood gradient is

$$\begin{aligned} &\nabla_{\boldsymbol{\theta}} \mathbb{E}_{p_d(\mathbf{x}, \mathbf{z})} \log p_{\boldsymbol{\theta}}(\mathbf{x}, \mathbf{z}) \\ &= \mathbb{E}_{p_d(\mathbf{x}, \mathbf{z})} \nabla_{\boldsymbol{\theta}} f_{\boldsymbol{\theta}}(\mathbf{x}, \mathbf{z}) - \mathbb{E}_{p_{\boldsymbol{\theta}}(\hat{\mathbf{x}}, \hat{\mathbf{z}})} \nabla_{\boldsymbol{\theta}} f_{\boldsymbol{\theta}}(\hat{\mathbf{x}}, \hat{\mathbf{z}}). \end{aligned} \qquad (4)$$

In contrastive to P-CEM that incorporates parametric cluster centers, the joint distribution of NP-CEM (Non-Parametric CEM) is directly defined based on the similarity between *instances* (Oord et al., 2018). We define the joint data distribution $p_d(\mathbf{x}, \mathbf{z}) = p_d(\mathbf{x}) p_d(\mathbf{z}|\mathbf{x})$ through re-parameterization,

$$\mathbf{z} = f_{\boldsymbol{\theta}}(t(\mathbf{x})), \quad \text{where } t \sim \mathcal{T} \text{ and } \mathbf{x} \sim p_d(\mathbf{x}), \qquad (5)$$

where $\mathcal{T}$ refers to the user-define pretext, e.g., a set of data augmentation operators $\mathcal{T} = \{t : \mathbb{R}^n \to \mathbb{R}^n\}$ [1].

## 3. Rediscovering Adversarial Training

In this section, we investigate why robust models have a good generative ability from the perspective of P-CEM. In general, AT solves the following *minimax* problem:

$$\min_{\boldsymbol{\theta}} \mathbb{E}_{p_d(\mathbf{x}, y)} \left[ \max_{\|\hat{\mathbf{x}} - \mathbf{x}\|_p \leq \varepsilon} \ell_{CE}(\hat{\mathbf{x}}, y; \boldsymbol{\theta}) \right]. \qquad (6)$$

---

[1]For the data pair, we detach the gradient of $\mathbf{z}$ w.r.t. $\boldsymbol{\theta}$ and assume that $\mathbf{x}$ and $t(\mathbf{x})$ have the same marginal distribution $p_d(\mathbf{x})$.

The *inner maximization* problem is to find an adversarial example $\hat{\mathbf{x}}$ within the $\ell_p$-norm $\varepsilon$-ball around the natural example $\mathbf{x}$ that maximizes the CE loss. While the *outer minimization* problem is to find model parameters that minimize the loss on the adversarial examples $\hat{\mathbf{x}}$.

**Maximization Process.** For the inner maximization problem, Projected Gradient Descent (PGD) (Madry et al., 2018) is the commonly used method, which generates the adversarial example $\hat{\mathbf{x}}$ by maximizing the CE loss[2] starting from $\hat{\mathbf{x}}_0 = \mathbf{x}$, that is,

$$\hat{\mathbf{x}}_{n+1} = \hat{\mathbf{x}}_n + \alpha \nabla_{\hat{\mathbf{x}}_n} \ell(\hat{\mathbf{x}}_n, y; \boldsymbol{\theta}) = \hat{\mathbf{x}}_n - \alpha \nabla_{\hat{\mathbf{x}}_n} \log p_{\boldsymbol{\theta}}(y|\hat{\mathbf{x}}_n)$$

$$= \hat{\mathbf{x}}_n + \alpha \nabla_{\hat{\mathbf{x}}_n} \left[ \log \sum_{k=1}^{K} \exp(f_{\boldsymbol{\theta}}(\hat{\mathbf{x}}_n, k)) \right] - \alpha \nabla_{\hat{\mathbf{x}}_n} f_{\boldsymbol{\theta}}(\hat{\mathbf{x}}_n, y),$$

while the Langevin dynamics for sampling from P-CEM starts from random noise $\hat{\mathbf{x}}_0 = \boldsymbol{\delta}$ and updates with

$$\hat{\mathbf{x}}_{n+1} = \hat{\mathbf{x}}_n + \alpha \nabla_{\hat{\mathbf{x}}} \log p_{\boldsymbol{\theta}}(\hat{\mathbf{x}}_n) + \sqrt{2\alpha} \cdot \boldsymbol{\varepsilon}$$

$$= \hat{\mathbf{x}}_n + \alpha \nabla_{\hat{\mathbf{x}}_n} \left[ \log \sum_{k=1}^{K} \exp(f_{\boldsymbol{\theta}}(\hat{\mathbf{x}}_n, k)) \right] + \sqrt{2\alpha} \cdot \boldsymbol{\varepsilon}.$$

The two update rules above both have a positive logsumexp gradient (the second term) to push up the marginal probability $p_{\boldsymbol{\theta}}(\hat{\mathbf{x}})$. As for the third term, PGD starts from a data point $(\mathbf{x}, y)$ such that it requires the repulsive gradient to be away from the original data point and do the exploration in a local region. Langevin dynamics instead starts from a random noise and an additive noise $\boldsymbol{\varepsilon}$ is injected for exploration. Thus, the maximization process in AT can be seen as a (biased) sampling method that draws samples from the corresponding probabilistic model $p_{\boldsymbol{\theta}}(\hat{\mathbf{x}})$. Compared to Langevin dynamics, PGD imposes specific inductive bias for sampling. With the additional repulsive gradient and $\varepsilon$-ball constraint, it explicitly encourages the samples to be misclassified around the original data. In practice, adversarial training with such adversarial examples is generally more stable than training P-CEM with Langevin samples, which indicates that PGD attack is a competitive alternative for the negative sampling method for P-CEM training.

**Minimization Process.** To begin with, the gradient of the joint log likelihood for P-CEM can be written as:

$$\begin{aligned} &\nabla_{\boldsymbol{\theta}} \mathbb{E}_{p_d(\mathbf{x}, y)} \log p_{\boldsymbol{\theta}}(\mathbf{x}, y) \\ &= \mathbb{E}_{p_d(\mathbf{x}, y)} \nabla_{\boldsymbol{\theta}} f_{\boldsymbol{\theta}}(\mathbf{x}, y) - \mathbb{E}_{p_{\boldsymbol{\theta}}(\hat{\mathbf{x}}, \hat{y})} \nabla_{\boldsymbol{\theta}} f_{\boldsymbol{\theta}}(\hat{\mathbf{x}}, \hat{y}) \\ &= \mathbb{E}_{p_d(\mathbf{x}, y)} \nabla_{\boldsymbol{\theta}} f_{\boldsymbol{\theta}}(\mathbf{x}, y) - \mathbb{E}_{p_{\boldsymbol{\theta}}(\hat{\mathbf{x}}) p_{\boldsymbol{\theta}}(\hat{y}|\hat{\mathbf{x}})} \nabla_{\boldsymbol{\theta}} f_{\boldsymbol{\theta}}(\hat{\mathbf{x}}, \hat{y}), \end{aligned}$$

where $(\mathbf{x}, y) \sim p_d(\mathbf{x}, y)$ denotes the positive data pair, and $(\hat{\mathbf{x}}, \hat{y}) \sim p_{\boldsymbol{\theta}}(\hat{\mathbf{x}}, \hat{y})$ denotes the negative sample pair. As discussed above, the adversarial examples $\hat{\mathbf{x}}$ generated by the maximization process can be regarded as negative samples,

---

[2]Note that we omit the projection operation and the gradient re-normalization steps for simplicity.

and $\hat{y} \sim p_{\boldsymbol{\theta}}(\hat{y}|\hat{\mathbf{x}})$ denotes the predicted label of $\hat{\mathbf{x}}$. To see how the maximum likelihood training of P-CEM is related to the minimization process of AT, we add an interpolated adversarial pair $(\hat{\mathbf{x}}, y)$ into Eqn. (3) and decompose it as the consistency gradient and the contrastive gradient:

$$
\begin{aligned}
&\nabla_{\boldsymbol{\theta}} \, \mathbb{E}_{p_d(\mathbf{x},y)} \log p_{\boldsymbol{\theta}}(\mathbf{x},y) \\
&= \mathbb{E}_{p_d(\mathbf{x},y) \otimes p_{\boldsymbol{\theta}}(\hat{\mathbf{x}},\hat{y})} [\nabla_{\boldsymbol{\theta}} f_{\boldsymbol{\theta}}(\mathbf{x},y) - \nabla_{\boldsymbol{\theta}} f_{\boldsymbol{\theta}}(\hat{\mathbf{x}},\hat{y})] \\
&= \mathbb{E}_{p_d(\mathbf{x},y) \otimes p_{\boldsymbol{\theta}}(\hat{\mathbf{x}},\hat{y})} \big[ \underbrace{\nabla_{\boldsymbol{\theta}} f_{\boldsymbol{\theta}}(\mathbf{x},y) - \nabla_{\boldsymbol{\theta}} f_{\boldsymbol{\theta}}(\hat{\mathbf{x}},y)}_{\text{consistency gradient}} \\
&\qquad\qquad + \underbrace{\nabla_{\boldsymbol{\theta}} f_{\boldsymbol{\theta}}(\hat{\mathbf{x}},y) - \nabla_{\boldsymbol{\theta}} f_{\boldsymbol{\theta}}(\hat{\mathbf{x}},\hat{y})}_{\text{contrastive gradient}} \big].
\end{aligned}
$$

Next, we show that the two parts correspond to two effective mechanisms developed in the adversarial training literature.

**AT loss.** As the two sample pairs in the contrastive gradient share the same input $\hat{\mathbf{x}}$, we can see that the contrastive gradient (second term) can be written equivalently as

$$
\begin{aligned}
&\mathbb{E}_{p_d(\mathbf{x},y) \otimes p_{\boldsymbol{\theta}}(\hat{\mathbf{x}},\hat{y})} [\nabla_{\boldsymbol{\theta}} f_{\boldsymbol{\theta}}(\hat{\mathbf{x}},y) - \nabla_{\boldsymbol{\theta}} f_{\boldsymbol{\theta}}(\hat{\mathbf{x}},\hat{y})] \\
&= \mathbb{E}_{p_d(\mathbf{x},y) \otimes p_{\boldsymbol{\theta}}(\hat{\mathbf{x}})} \big[ \nabla_{\boldsymbol{\theta}} f_{\boldsymbol{\theta}}(\hat{\mathbf{x}},y) - \mathbb{E}_{p_{\boldsymbol{\theta}}(\hat{y}|\hat{\mathbf{x}})} \nabla_{\boldsymbol{\theta}} f_{\boldsymbol{\theta}}(\hat{\mathbf{x}},\hat{y}) \big] \\
&= \mathbb{E}_{p_d(\mathbf{x},y) \otimes p_{\boldsymbol{\theta}}(\hat{\mathbf{x}})} [\nabla_{\boldsymbol{\theta}} \log p_{\boldsymbol{\theta}}(y|\hat{\mathbf{x}})],
\end{aligned}
$$

which is exactly the negative gradient of the AT loss $\ell_{CE}(\hat{\mathbf{x}}, y; \boldsymbol{\theta})$ in Eqn. (6) (see details in Appendix).

**Regularization.** As for the consistency gradient (first term), original AT (Madry et al., 2018) simply ignores it. Its variant TRADES (Zhang et al., 2019) instead proposes the KL regularization $\mathrm{KL}(p(\cdot|\hat{\mathbf{x}})\|p(\cdot|\mathbf{x}))$ that regularizes the consistency of the predicted probabilities on all classes, whose optimum implies that the consistency gradient vanishes, i.e., $p(\cdot|\hat{\mathbf{x}}) = p(\cdot|\mathbf{x}) \to f_{\boldsymbol{\theta}}(\mathbf{x},y) = f_{\boldsymbol{\theta}}(\hat{\mathbf{x}},y)$.

The above analysis indicates that the minimization objective of AT is closely related to the maximum likelihood training of P-CEM, and TRADES further injects adversarial robustness prior by regularizing the consistency gradient. Together with the analysis on the maximization process, we show that AT is a competitive alternative for training P-CEM (a generative model) with more stable training behaviors. That explains why robust models with AT are also generative.

### 3.1. Discussion on Standard Training

In the above discussion, we have explained why adversarial training is generative from the perspective of P-CEM. In fact, it can also help characterize why classifiers with Standard Training (ST) are not generative (i.e., poor sample quality). A key insight is that if we replace the model distribution $p_{\boldsymbol{\theta}}(\hat{\mathbf{x}})$ with the data distribution $p_d(\mathbf{x})$ in Eqn. (3), we have

$$
\begin{aligned}
&\nabla_{\boldsymbol{\theta}} \mathbb{E}_{p_d(\mathbf{x},y)} \log p_{\boldsymbol{\theta}}(\mathbf{x},y) \\
&= \mathbb{E}_{p_d(\mathbf{x},y)} \nabla_{\boldsymbol{\theta}} f_{\boldsymbol{\theta}}(\mathbf{x},y) - \mathbb{E}_{p_{\boldsymbol{\theta}}(\hat{\mathbf{x}}) p(\hat{y}|\hat{\mathbf{x}})} \nabla_{\boldsymbol{\theta}} f_{\boldsymbol{\theta}}(\hat{\mathbf{x}},\hat{y}) \\
&\approx \mathbb{E}_{p_d(\mathbf{x},y)} \nabla_{\boldsymbol{\theta}} f_{\boldsymbol{\theta}}(\mathbf{x},y) - \mathbb{E}_{p_d(\mathbf{x}) p(\hat{y}|\mathbf{x})} \nabla_{\boldsymbol{\theta}} f_{\boldsymbol{\theta}}(\mathbf{x},\hat{y}) \\
&= \nabla_{\boldsymbol{\theta}} \mathbb{E}_{p_d(\mathbf{x},y)} \log p_{\boldsymbol{\theta}}(y|\mathbf{x}),
\end{aligned}
$$

which is the negative gradient of the CE loss. Thus, ST is equivalent to training P-CEM by simply replacing model-based negative samples $\hat{\mathbf{x}} \sim p_{\boldsymbol{\theta}}(\mathbf{x})$ with data samples $\mathbf{x} \sim p_d(\mathbf{x})$. This approximation makes ST computationally efficient with good accuracy on natural data, but significantly limits its robustness on adversarial examples (as model-based negative samples). Similarly, because ST ignores exploring negative samples while training, standard classifiers also fail to generate realistic samples.

### 3.2. Extension to Unsupervised AT

In the above discussion, we establish the connection between AT and CEM in the supervised scenarios. Moreover, this perspective also enables us to generalize adversarial training to unsupervised scenarios via our unified framework, CEM. Specifically, we can extend the discussion of P-CEM (supervised) to NP-CEM (unsupervised) and derive the corresponding inner and outer optimization objectives in a principled way. Due to the limit of space, more details are provided in the Appendix.

## 4. Principled Adversarial Sampling

The interpretation of AT through our unified framework also inspires us to design principled sampling algorithms from robust models. Below, we present our adversarial sampling algorithms for both supervised and unsupervised scenarios.

### 4.1. Supervised Scenario

**Targeted Attack (TA).** Previously, to draw samples from a robust classifier, Santurkar et al. (2019) utilize a targeted attack that optimizes the input initialized by a random noise $\hat{\mathbf{x}}_0$ towards a specified class $\hat{y} \sim p_d(\hat{y})$:

$$
\begin{aligned}
\hat{\mathbf{x}}_{n+1} &= \hat{\mathbf{x}}_n + \alpha \nabla_{\mathbf{x}_n} \log p_{\boldsymbol{\theta}}(\hat{y}|\hat{\mathbf{x}}_n) \qquad (7) \\
&= \hat{\mathbf{x}}_n + \alpha \nabla_{\mathbf{x}} f(\hat{\mathbf{x}}_n, \hat{y}) - \alpha \nabla_{\hat{\mathbf{x}}_n} \left[ \log \sum_{k=1}^{K} \exp(f_{\boldsymbol{\theta}}(\hat{\mathbf{x}}_n, k)) \right].
\end{aligned}
$$

Compared to PGD attack in Eqn. (**??**), while pushing $\hat{\mathbf{x}}$ towards $\hat{y}$, TA has a negative $\mathrm{logsumexp}$ gradient that decreases the marginal probability $p_{\boldsymbol{\theta}}(\hat{\mathbf{x}})$. This could explain why TA is less powerful for adversarial attack and is rarely used for adversarial training.

**Conditional Sampling (CS).** To overcome the drawback of targeted attack, a natural idea would be dropping the negative $\mathrm{logsumexp}$ gradient. In fact, we can show that this is equivalent to sampling from the conditional distribution:

$$
p_{\boldsymbol{\theta}}(\mathbf{x}|\hat{y}) = \frac{\exp(f_{\boldsymbol{\theta}}(\mathbf{x},\hat{y}))}{Z_{\mathbf{x}|\hat{y}}(\boldsymbol{\theta})}, \ \ Z_{\mathbf{x}|\hat{y}}(\boldsymbol{\theta}) = \int_{\mathbf{x}} \exp(f_{\boldsymbol{\theta}}(\mathbf{x},\hat{y})) d\mathbf{x},
$$

and its Langevin dynamics takes the form:

$$
\begin{aligned}
\hat{\mathbf{x}}_{n+1} &= \mathbf{x}_n + \alpha \nabla_{\hat{\mathbf{x}}_n} \log p_{\boldsymbol{\theta}}(\hat{\mathbf{x}}_n|\hat{y}) + \sqrt{2\alpha} \cdot \boldsymbol{\varepsilon} \\
&= \hat{\mathbf{x}}_n + \alpha \nabla_{\hat{\mathbf{x}}_n} f_{\boldsymbol{\theta}}(\hat{\mathbf{x}}_n, \hat{y}) + \sqrt{2\alpha} \cdot \boldsymbol{\varepsilon}.
\end{aligned}
$$

Table 1. Inception Scores (IS) and Fréchet Inception Distance (FID) of different generative models. Results marked with ⋆ are taken from Shmelkov et al. (2018).

| Method | IS (↑) | FID (↓) |
|---|---|---|
| **Auto-regressive** | | |
| PixelCNN++⋆ (Salimans et al., 2017) | 5.36 | 119.5 |
| **GAN-based** | | |
| DCGAN⋆ (Radford et al., 2016) | 6.69 | 35.6 |
| WGAN-GP (Gulrajani et al., 2017) | 7.86 | 36.4 |
| PresGAN (Dieng et al., 2019) | - | 52.2 |
| StyleGAN2-ADA (Karras et al., 2020) | **10.02** | - |
| **Score-based** | | |
| NCSN (Song & Ermon, 2019) | 8.87 | 25.32 |
| DDPM (Ho et al., 2020) | 9.46 | 3.17 |
| NCSN++ (Song et al., 2020) | 9.89 | **2.20** |
| **EBM-based** | | |
| P-CEM (Grathwohl et al., 2019) | 8.76 | 38.4 |
| DRL (Gao et al., 2021) | 8.58 | 9.60 |
| **AT-based** | | |
| TA (Santurkar et al., 2019) (w/ ResNet50) | 7.5 | - |
| **Supervised CEM** (w/ ResNet50) | **9.77** | 56.26 |
| **Unsupervised CEM** (w/ ResNet18) (ours) | 8.68 | **36.4** |
| **Unsupervised CEM** (w/ ResNet50) (ours) | 9.61 | 40.25 |

Table 2. Inception Scores (IS) and Fréchet Inception Distance (FID) of different sampling methods for adversarially robust models. Cond: conditional. Uncond: unconditional.

| Training | Sampling | Method | IS (↑) | FID (↓) |
|---|---|---|---|---|
| Supervised | Cond | TA | 9.26 | 56.72 |
| | | Langevin | 9.65 | 63.34 |
| | | CS | 9.77 | 56.26 |
| Unsupervised (w/ ResNet18) | Uncond | PGD | 5.35 | 74.27 |
| | | Langevin | **8.24** | **41.80** |
| | Cond | PGD | 5.85 | 68.54 |
| | | Langevin | **8.68** | **36.44** |
| Unsupervised (w/ ResNet50) | Uncond | PGD | 5.24 | 141.54 |
| | | Langevin | **9.57** | **44.86** |
| | Cond | PGD | 5.37 | 137.68 |
| | | Langevin | **9.61** | **40.25** |

Samples drawn this way essentially follow an approximated model distribution, $p_{\boldsymbol{\theta}}(\hat{\mathbf{x}}, \hat{y}) \approx p_d(\hat{y})p_{\boldsymbol{\theta}}(\hat{\mathbf{x}}|\hat{y})$. Thus, CS can be seen as a debiased targeted attack algorithm.

### 4.2. Unsupervised Scenario

Without labels, we can also draw samples from $p_{\boldsymbol{\theta}}(\mathbf{x})$ from our unsupervised robust models, i.e., NP-CEM, by performing Langevin dynamics with $K$ negative samples $\{\mathbf{z}_k^-\}$:

$$\hat{\mathbf{x}}_{n+1} = \hat{\mathbf{x}}_n + \alpha\nabla_{\hat{\mathbf{x}}_n}\log p_{\boldsymbol{\theta}}(\hat{\mathbf{x}}_n) + \sqrt{2\alpha}\cdot\boldsymbol{\varepsilon}$$

$$\approx \hat{\mathbf{x}}_n + \alpha\nabla_{\hat{\mathbf{x}}_n}\left[\log\frac{1}{K}\sum_{k=1}^{K}p_{\boldsymbol{\theta}}(\hat{\mathbf{x}}_n, \mathbf{z}_k^-)\right] + \sqrt{2\alpha}\cdot\boldsymbol{\varepsilon}$$

$$= \hat{\mathbf{x}}_n + \alpha\nabla_{\hat{\mathbf{x}}_n}\left[\log\sum_{k=1}^{K}\exp(f_{\boldsymbol{\theta}}(\hat{\mathbf{x}}_n, \mathbf{z}_k^-))\right] + \sqrt{2\alpha}\cdot\boldsymbol{\varepsilon}.$$

In comparison, as we do not have access to real data points as our anchor points for adversarial attack, PGD attack may not be effective at sampling for the unsupervised scenario.

## 5. Experiments

In this section, we validate our proposed method by evaluating the adversarial sampling algorithms derived from our framework. Specifically, we evaluate the sample equality with both supervised and unsupervised robust models on CIFAR-10. For supervised robust models, we adopt the same pretrained ResNet50 checkpoint on CIFAR-10 as San-

turkar et al. (2019). As for the unsupervised case, we are the first to consider sampling from unsupervised robust models. We train ResNet18 and ResNet50 (He et al., 2016) encoders following the setup of an existing unsupervised adversarial training method ACL (Jiang et al., 2020). The training attack is kept the same as that of the supervised case for a fair comparison. As for the sampling algorithms, we tune the step size $\alpha$, noise scale $\varepsilon$, and number of steps $K$ for better sample quality. More details are provided in Appendix.

**Comparison with other generative models**. In Table 1, we compare the sample quality of adversarial sampling methods with different kinds of generative models, where our adversarial sampling methods outperform many deep generative models and obtain state-of-the-art Inception scores on par with StyleGAN2 (Karras et al., 2020).

**Comparison among adversarial sampling methods.** In Table 2, we further compare the sample quality of different adversarial sampling methods. For supervised models, we can see that indeed TA obtains the lowest IS, while CS can significantly refine the sample quality. For unsupervised models, we can see that Langevin dynamics outperforms PGD consistently by a large margin. In particular, conditional sampling initialized with class-wise noise can improve a little on the sample quality compared to unconditional sampling.

## 6. Conclusion

In this paper, we proposed a unified probabilistic framework, named Contrastive Energy-based Model (CEM), which not only explains the generative ability of adversarial training, but also provides a unified perspective of adversarial training and sampling in both supervised and unsupervised paradigms. Extensive experiments show that sampling methods derived from our framework indeed demonstrate better robustness and sample quality than state-of-the-art methods.

## Acknowledgement

Yisen Wang is supported by the National Natural Science Foundation of China under Grant No. 62006153 and Project 2020BD006 supported by PKU-Baidu Fund. Zhouchen Lin is supported by the National Natural Science Foundation of China (Grant No.s 61625301 and 61731018), Project 2020BD006 supported by PKU-Baidu Fund, Major Scientific Research Project of Zhejiang Lab (Grant No.s 2019KB0AC01 and 2019KB0AB02), and Beijing Academy of Artificial Intelligence. Jiansheng Yang is supported by the National Science Foundation of China under Grant No. 11961141007.

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

# A. Principled Unsupervised Adversarial Training with CEM

In this section, we show that with our unified framework, we can naturally extend the interpretation developed for supervised adversarial training to the unsupervised scenario.

## A.1. Understanding Unsupervised Standard Training

Recently, the following InfoNCE loss is widely adopted for unsupervised contrastive learning of data representations (Oord et al., 2018; Chen et al., 2020; He et al., 2020),

$$\ell_{NCE}(\mathbf{x}, \bar{\mathbf{x}}, \hat{\mathcal{X}}; \boldsymbol{\theta}) = -\log \frac{\exp(f_{\boldsymbol{\theta}}(\mathbf{x}, \bar{\mathbf{x}}))}{\sum_{i=j}^{K} \exp(f_{\boldsymbol{\theta}}(\mathbf{x}, \hat{\mathbf{x}}_j))}, \quad (8)$$

where $f_{\boldsymbol{\theta}}(\mathbf{x}, \hat{\mathbf{x}}) = g_{\boldsymbol{\theta}}(\mathbf{x})^{\top} g_{\boldsymbol{\theta}}(\hat{\mathbf{x}})$ calculates the similarity between the representations of the two data samples, $\mathbf{x}, \bar{\mathbf{x}}$ are generated by two random augmentations (drawn from $\mathcal{T}$) of the same data example, and $\hat{\mathcal{X}} = \{\hat{\mathbf{x}}_j\}_{j=1}^{K}$ consists of $K$ independently drawn negative samples. In practice, one of the $K$ negative samples is chosen to be the positive sample $\bar{\mathbf{x}}$. Therefore, InfoNCE can be seen as an instance-wise $K$-class cross entropy loss for non-parametric classification.

Perhaps surprisingly, we show that the InfoNCE loss is equivalent to the importance sampling estimate of our NP-CEM by approximating the negative samples from $p_{\boldsymbol{\theta}}(\mathbf{x})$ with data samples from $p_d(\mathbf{x})$, as what we have done in standard supervised training (Section C.3):

$$\mathbb{E}_{p_d(\mathbf{x}, \mathbf{z})} \nabla_{\boldsymbol{\theta}} f_{\boldsymbol{\theta}}(\mathbf{x}, \mathbf{z}) - \mathbb{E}_{p_{\boldsymbol{\theta}}(\hat{\mathbf{x}}, \hat{\mathbf{z}})} \nabla_{\boldsymbol{\theta}} f_{\boldsymbol{\theta}}(\hat{\mathbf{x}}, \hat{\mathbf{z}})$$

$$= \mathbb{E}_{p_d(\mathbf{x}, \mathbf{z})} \nabla_{\boldsymbol{\theta}} f_{\boldsymbol{\theta}}(\mathbf{x}, \mathbf{z}) - \mathbb{E}_{p_{\boldsymbol{\theta}}(\hat{\mathbf{x}}) p_{\boldsymbol{\theta}}(\hat{\mathbf{z}})} \frac{p_{\boldsymbol{\theta}}(\hat{\mathbf{z}}|\hat{\mathbf{x}})}{p_{\boldsymbol{\theta}}(\hat{\mathbf{z}})} \nabla_{\boldsymbol{\theta}} f_{\boldsymbol{\theta}}(\hat{\mathbf{x}}, \hat{\mathbf{z}})$$

$$\approx \mathbb{E}_{p_d(\mathbf{x}, \mathbf{z})} \nabla_{\boldsymbol{\theta}} f_{\boldsymbol{\theta}}(\mathbf{x}, \mathbf{z}) - \mathbb{E}_{p_d(\hat{\mathbf{x}}) p_d(\hat{\mathbf{z}})} \frac{p_{\boldsymbol{\theta}}(\hat{\mathbf{z}}|\hat{\mathbf{x}})}{p_{\boldsymbol{\theta}}(\hat{\mathbf{z}})} \nabla_{\boldsymbol{\theta}} f_{\boldsymbol{\theta}}(\hat{\mathbf{x}}, \hat{\mathbf{z}})$$

$$\approx \frac{1}{N} \sum_{i=1}^{N} \nabla_{\boldsymbol{\theta}} \log \frac{\exp(f_{\boldsymbol{\theta}}(\mathbf{x}_i, \mathbf{z}_i))}{\sum_{k=1}^{K} \exp(f_{\boldsymbol{\theta}}(\mathbf{x}_i, \mathbf{z}_{ik}^{-}))}, \quad (9)$$

which is exactly the negative gradient of the InfoNCE loss. In the above analysis, for an empirical estimate, we draw $N$ positive pairs $(\mathbf{x}_i, \mathbf{z}_i) \sim p_d(\mathbf{x}, \mathbf{z})$, and for each anchor $\mathbf{x}_i$, we further draw $K$ negative samples $\{\mathbf{z}_{ik}^{-}\}$ from $p_d(\hat{\mathbf{z}})$ through reparameterization (Eqn. (5)).

The negative phase of NP-CEM is supposed to sample from $p_{\boldsymbol{\theta}}(\hat{\mathbf{z}}|\hat{\mathbf{x}})$, where samples semantically close to the anchor sample $\hat{\mathbf{x}}$, a.k.a. hard negative samples, should have high probabilities. However, InfoNCE adopts a *non-informative* uniform proposal $p_d(\hat{\mathbf{z}})$ for importance sampling, which is very sample inefficient because most samples are useless (Kalantidis et al., 2020). This observation motivates us to design more efficient sampling scheme for contrastive learning by mining hard negatives. For example, Robinson et al. (2021) directly replace the plain proposal with $\tilde{p}_{\boldsymbol{\theta}}(\hat{\mathbf{z}}|\hat{\mathbf{x}}) = \frac{\exp(\beta f_{\boldsymbol{\theta}}(\hat{\mathbf{x}}, \hat{\mathbf{z}}))}{Z_{\beta}(\boldsymbol{\theta})}$ while keeping the reweighing term. From the perspective of NP-CEM, it will introduce bias into

the importance sampling, which should be treated carefully. In all, NP-CEM provides a principled framework to characterize and develop contrastive learning algorithms.

### A.2. Proposed Unsupervised Adversarial Training

AT is initially designed for supervised learning, where adversarial examples can be clearly defined by misclassification. However, it remain unclear what is the right way to do Unsupervised Adversarial Training (UAT) without access to *any* labels. Previous works (Jiang et al., 2020; Ho & Nvasconcelos, 2020; Kim et al., 2020) have carried out UAT with the adversarial InfoNCE loss, which works well but lacks theoretical justification. Our unified CEM framework offers a principled way to generalize adversarial training from supervised to unsupervised scenarios.

**Maximization Process.** Sampling from $p_{\boldsymbol{\theta}}(\mathbf{x})$ can be more difficult than that in supervised scenarios because it does not admit a closed form for variable $\mathbf{z}$. Thus, we perform Langevin dynamics with $K$ negative samples $\{\mathbf{z}_k^-\}$,

$$\hat{\mathbf{x}}_{n+1} = \hat{\mathbf{x}}_n + \alpha \nabla_{\hat{\mathbf{x}}_n} \log p_{\boldsymbol{\theta}}(\hat{\mathbf{x}}_n) + \sqrt{2\alpha} \cdot \boldsymbol{\varepsilon} \quad (10)$$

$$\approx \hat{\mathbf{x}}_n + \alpha \nabla_{\hat{\mathbf{x}}_n} \left[ \log \frac{1}{K} \sum_{k=1}^{K} p_{\boldsymbol{\theta}}(\hat{\mathbf{x}}_n, \mathbf{z}_k^-) \right] + \sqrt{2\alpha} \cdot \boldsymbol{\varepsilon}$$

$$= \hat{\mathbf{x}}_n + \alpha \nabla_{\hat{\mathbf{x}}_n} \left[ \log \sum_{k=1}^{K} \exp(f_{\boldsymbol{\theta}}(\hat{\mathbf{x}}_n, \mathbf{z}_k^-)) \right] + \sqrt{2\alpha} \cdot \boldsymbol{\varepsilon}.$$

While the PGD attack of the InfoNCE loss (Eqn. 18),

$$\hat{\mathbf{x}}_{n+1} = \hat{\mathbf{x}}_n + \alpha \nabla_{\hat{\mathbf{x}}_n} \log \frac{\exp(f_{\boldsymbol{\theta}}(\hat{\mathbf{x}}_n, \mathbf{z}))}{\sum_{k=1}^{K} \exp(f_{\boldsymbol{\theta}}(\hat{\mathbf{x}}_n, \mathbf{z}_k^-))} \quad (11)$$

$$= \hat{\mathbf{x}}_n + \alpha \nabla_{\hat{\mathbf{x}}_n} \left[ \log \sum_{k=1}^{K} \exp(f_{\boldsymbol{\theta}}(\hat{\mathbf{x}}_n, \mathbf{z}_k^-)) \right] - \alpha \nabla_{\boldsymbol{\theta}} f_{\boldsymbol{\theta}}(\hat{\mathbf{x}}_n, \mathbf{z}),$$

resembles the Langevin dynamics as they both share the positive logsumexp gradient that pushes up $p_{\boldsymbol{\theta}}(\hat{\mathbf{x}})$, and differs by a repulse negative gradient $-f_{\boldsymbol{\theta}}(\hat{\mathbf{x}}, \mathbf{z})$ away from the anchor $\mathbf{z}$, which is a direct analogy of the PGD attack in supervised learning (Section 3). Therefore, we believe that the PGD attack of InfoNCE is a proper way to craft adversarial examples by sampling from $p_{\boldsymbol{\theta}}(\mathbf{x})$.

**Minimization Process.** Following the same routine in Section C.2, with the adversarial example $\hat{\mathbf{x}} \sim p_{\boldsymbol{\theta}}(\hat{\mathbf{x}})$, we can insert an interpolated adversarial pair $(\hat{\mathbf{x}}, \mathbf{z})$ and decompose the gradient of NP-CEM into the consistency gradient and the contrastive gradient,

$$\nabla_{\boldsymbol{\theta}} \mathbb{E}_{p_d(\mathbf{x}, \mathbf{z})} \log p_{\boldsymbol{\theta}}(\mathbf{x}, \mathbf{z})$$

$$= \mathbb{E}_{p_d(\mathbf{x}, \mathbf{z}) \otimes p_{\boldsymbol{\theta}}(\hat{\mathbf{x}}, \hat{\mathbf{z}})} [\nabla_{\boldsymbol{\theta}} f_{\boldsymbol{\theta}}(\mathbf{x}, \mathbf{z}) - \nabla_{\boldsymbol{\theta}} f_{\boldsymbol{\theta}}(\hat{\mathbf{x}}, \mathbf{z})]$$

$$= \mathbb{E}_{p_d(\mathbf{x}, \mathbf{z}) \otimes p_{\boldsymbol{\theta}}(\hat{\mathbf{x}}, \hat{\mathbf{z}})} \Big[ \underbrace{\nabla_{\boldsymbol{\theta}} f_{\boldsymbol{\theta}}(\mathbf{x}, \mathbf{z}) - \nabla_{\boldsymbol{\theta}} f_{\boldsymbol{\theta}}(\hat{\mathbf{x}}, \mathbf{z})}_{\text{consistency gradient}} \quad (12)$$

$$+ \underbrace{\nabla_{\boldsymbol{\theta}} f_{\boldsymbol{\theta}}(\hat{\mathbf{x}}, \mathbf{z}) - \nabla_{\boldsymbol{\theta}} f_{\boldsymbol{\theta}}(\hat{\mathbf{x}}, \hat{\mathbf{z}})}_{\text{contrastive gradient}} \Big].$$

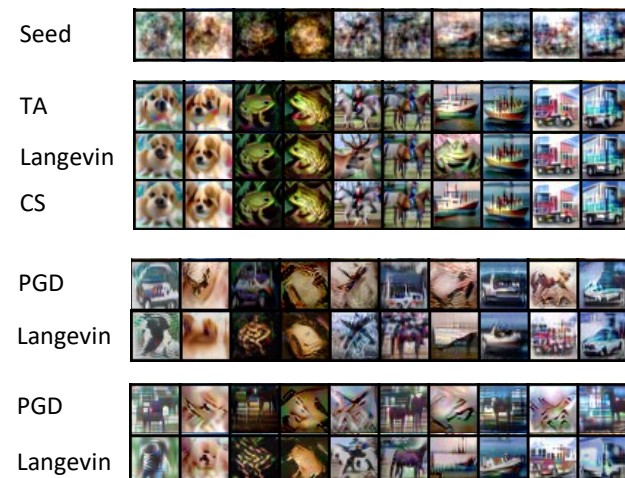

*Figure 1.* Randomly drawn samples with different sampling methods. Four groups of samples from top to bottom: initial seed, supervised models (ResNet50), unsupervised models (ResNet18), unsupervised models (ResNet50).

Following the analysis for supervised AT, it is also easy to see that the contrastive gradient is equivalent to the gradient of the Adversarial InfoNCE loss utilized in previous work (Jiang et al., 2020; Ho & Nvasconcelos, 2020; Kim et al., 2020) with adversarial example $\hat{\mathbf{x}}$,

$$\mathbb{E}_{p_d(\mathbf{x}, \mathbf{z}) \otimes p_{\boldsymbol{\theta}}(\hat{\mathbf{x}}, \hat{\mathbf{z}})} [\nabla_{\boldsymbol{\theta}} f_{\boldsymbol{\theta}}(\hat{\mathbf{x}}, \mathbf{z}) - \nabla_{\boldsymbol{\theta}} f_{\boldsymbol{\theta}}(\hat{\mathbf{x}}, \hat{\mathbf{z}})]$$

$$= \mathbb{E}_{p_d(\mathbf{x}, \mathbf{z}) \otimes p_{\boldsymbol{\theta}}(\hat{\mathbf{x}})} \left[ \nabla_{\boldsymbol{\theta}} f_{\boldsymbol{\theta}}(\hat{\mathbf{x}}, \mathbf{z}) - \mathbb{E}_{p_{\boldsymbol{\theta}}(\hat{\mathbf{z}}|\hat{\mathbf{x}})} \nabla_{\boldsymbol{\theta}} f_{\boldsymbol{\theta}}(\hat{\mathbf{x}}, \hat{\mathbf{z}}) \right]$$

$$\approx \mathbb{E}_{p_d(\mathbf{x}, \mathbf{z}) \otimes p_{\boldsymbol{\theta}}(\hat{\mathbf{x}})} \left[ \nabla_{\boldsymbol{\theta}} f_{\boldsymbol{\theta}}(\hat{\mathbf{x}}, \mathbf{z}) - \mathbb{E}_{p_d(\hat{\mathbf{z}})} \frac{p_{\boldsymbol{\theta}}(\hat{\mathbf{z}}|\hat{\mathbf{x}})}{p_{\boldsymbol{\theta}}(\hat{\mathbf{z}})} \nabla_{\boldsymbol{\theta}} f_{\boldsymbol{\theta}}(\hat{\mathbf{x}}, \hat{\mathbf{z}}) \right]$$

$$\approx \frac{1}{N} \sum_{i=1}^{N} \nabla_{\boldsymbol{\theta}} \log \frac{\exp(f_{\boldsymbol{\theta}}(\hat{\mathbf{x}}_i, \mathbf{z}_i))}{\sum_{k=1}^{K} \exp(f_{\boldsymbol{\theta}}(\hat{\mathbf{x}}_i, \mathbf{z}_{ik}^-))}. \quad (13)$$

## B. More Details on Adversarial Sampling

In this part, we provide more details of our adversarial sampling experiments.

**Models.** For supervised robust models, we adopt the same pretrained ResNet50 checkpoint on CIFAR-10 as Santurkar et al. (2019) [3] for a fair comparison. The model is adversarially trained with $\ell_2$-norm PGD attack with random start, maximal perturbation norm 0.5, step size 0.1 and 7 steps. As for the unsupervised case, we are the first to consider sampling from unsupervised robust models. We train ResNet18 and ResNet50 (He et al., 2016) encoders following the setup of an existing unsupervised adversarial training method ACL (Jiang et al., 2020). The training attack is kept the same as that of the supervised case for a fair comparison. More details are provided in Appendix.

---

[3] We download the checkpoint from the repository https://github.com/MadryLab/robustness_applications.

**Sampling protocol.** In practice, our adversarial sampling methods take the following general form as a mixture of the PGD and Langevin dynamics,

$$\mathbf{x}_{n+1} = \Pi_{\|\mathbf{x}_n - \mathbf{x}_0\|_2 \leq \beta} \left[ \mathbf{x}_n + \alpha \mathbf{g}_k + \eta \varepsilon_k \right],$$
$$\mathbf{x}_0 = \boldsymbol{\delta}, \varepsilon_k \sim \mathcal{N}(\mathbf{0}, \mathbf{1}), k = 0, 1, \ldots, K,$$

where $\mathbf{g}_k$ is the update gradient, $\varepsilon_k$ is the diffusion noise, $\Pi_{\mathcal{S}}$ is the projector operator, and $\boldsymbol{\delta}$ is the (conditional) initial seeds drawn from the multivariate normal distribution whose mean and covariance are calculated from the CIFAR-10 test set following Santurkar et al. (2019) (details in Appendix). Note that there are four hyper-parameters in our sampling protocol: step size $\alpha$, $\ell_2$-ball size $\beta$, noise scale $\eta$, and iteration steps $K$, for which we list our choice in Table **??** (the analysis on these parameters can be found in Appendix). We evaluate the sample quality quantitatively with Inception Score (IS) (Salimans et al., 2016) and Fréchet Inception Distance (FID) (Heusel et al., 2017).

**Visualization.** We also demonstrate our randomly drawn (not cherry-picking) samples in Figure 1. It can be seen that both our supervised and unsupervised methods achieve significantly better sample quality than previous baselines.

# C. Omitted Technical Details

For a concise presentation, we have omitted several technical details in the main text. Here we present a complete description of the derivation process.

## C.1. Log Likelihood Gradient of EBM

In Section 3, we have introduced Energy-based Models (EBM) and the gradient of their log likelihood. We now show how it can be derived out.

For a EBM in the following form,

$$p_{\boldsymbol{\theta}}(\mathbf{x}) = \frac{\exp\left(-E_{\boldsymbol{\theta}}(\mathbf{x})\right)}{Z(\boldsymbol{\theta})}, \tag{14}$$

the gradient of the log likelihood can be derived as follows:

$$\nabla_{\boldsymbol{\theta}} \mathbb{E}_{p_d(\mathbf{x})} \log p_{\boldsymbol{\theta}}(\mathbf{x}) \tag{15}$$
$$= -\mathbb{E}_{p_d(\mathbf{x})} \nabla_{\boldsymbol{\theta}} E_{\boldsymbol{\theta}}(\mathbf{x}) - \nabla_{\boldsymbol{\theta}} \log Z(\boldsymbol{\theta})$$
$$= \mathbb{E}_{p_d(\mathbf{x})} \nabla_{\boldsymbol{\theta}} E_{\boldsymbol{\theta}}(\mathbf{x}) + \frac{\nabla_{\boldsymbol{\theta}} \int_{\mathbf{x}} exp(-E_{\boldsymbol{\theta}}(\mathbf{x}))}{Z(\boldsymbol{\theta})}$$
$$= -\mathbb{E}_{p_d(\mathbf{x})} \nabla_{\boldsymbol{\theta}} E_{\boldsymbol{\theta}}(\mathbf{x}) + \int_{\hat{\mathbf{x}}} \frac{\exp(-E_{\boldsymbol{\theta}}(\hat{\mathbf{x}}))}{Z(\boldsymbol{\theta})} \nabla_{\boldsymbol{\theta}} E_{\boldsymbol{\theta}}(\hat{\mathbf{x}})$$
$$= \underbrace{-\mathbb{E}_{p_d(\mathbf{x})} \nabla_{\boldsymbol{\theta}} E_{\boldsymbol{\theta}}(\mathbf{x})}_{\text{positive phase}} + \underbrace{\mathbb{E}_{p_{\boldsymbol{\theta}}(\hat{\mathbf{x}})} \nabla_{\boldsymbol{\theta}} E_{\boldsymbol{\theta}}(\hat{\mathbf{x}})}_{\text{negative phase}}.$$

## C.2. Equivalence between AT Loss and Contrastive Gradient in Supervised Learning

In Section 3, we have claimed that the contrastive gradient equals to the negative gradient of the AT loss) following the

same deduction in Eqn. (17),

$$\mathbb{E}_{p_d(\mathbf{x}, y) \otimes p_{\boldsymbol{\theta}}(\hat{\mathbf{x}}, \hat{y})} \left[ \nabla_{\boldsymbol{\theta}} f_{\boldsymbol{\theta}}(\hat{\mathbf{x}}, y) - \nabla_{\boldsymbol{\theta}} f_{\boldsymbol{\theta}}(\hat{\mathbf{x}}, \hat{y}) \right]$$
$$= \mathbb{E}_{p_d(\mathbf{x}, y) \otimes p_{\boldsymbol{\theta}}(\hat{\mathbf{x}})} \left[ \nabla_{\boldsymbol{\theta}} f_{\boldsymbol{\theta}}(\hat{\mathbf{x}}, y) - \mathbb{E}_{p_{\boldsymbol{\theta}}(\hat{y}|\hat{\mathbf{x}})} \nabla_{\boldsymbol{\theta}} f_{\boldsymbol{\theta}}(\hat{\mathbf{x}}, \hat{y}) \right]$$
$$= \mathbb{E}_{p_d(\mathbf{x}, y)} \nabla_{\boldsymbol{\theta}} \log \frac{\exp(f_{\boldsymbol{\theta}}(\hat{\mathbf{x}}, y))}{\sum_{k=1}^{K} \exp(f_{\boldsymbol{\theta}}(\hat{\mathbf{x}}, k))}$$
$$= \mathbb{E}_{p_d(\mathbf{x}, y) \otimes p_{\boldsymbol{\theta}}(\hat{\mathbf{x}})} \nabla_{\boldsymbol{\theta}} \log p_{\boldsymbol{\theta}}(y|\hat{\mathbf{x}}), \tag{16}$$

which is exactly the negative gradient of the canonical AT loss (Madry et al., 2018).

## C.3. Connection between Standard Training and JEM

In Section 3.1, we have claimed that if we replace the model distribution $p_{\boldsymbol{\theta}}(\hat{\mathbf{x}})$ with the data distribution $p_d(\mathbf{x})$ in Eqn. (3), the log likelihood gradient of JEM is equivalent to the negative gradient of the CE loss. Here we give a detailed proof as follows:

$$\nabla_{\boldsymbol{\theta}} \mathbb{E}_{p_d(\mathbf{x}, y)} \log p_{\boldsymbol{\theta}}(\mathbf{x}, y)$$
$$= \mathbb{E}_{p_d(\mathbf{x}, y)} \nabla_{\boldsymbol{\theta}} f_{\boldsymbol{\theta}}(\mathbf{x}, y) - \mathbb{E}_{p_{\boldsymbol{\theta}}(\hat{\mathbf{x}}) p_{\boldsymbol{\theta}}(\hat{y}|\hat{\mathbf{x}})} \nabla_{\boldsymbol{\theta}} f_{\boldsymbol{\theta}}(\hat{\mathbf{x}}, \hat{y})$$
$$\approx \mathbb{E}_{p_d(\mathbf{x}, y)} \nabla_{\boldsymbol{\theta}} f_{\boldsymbol{\theta}}(\mathbf{x}, y) - \mathbb{E}_{p_d(\mathbf{x}) p_{\boldsymbol{\theta}}(\hat{y}|\mathbf{x})} \nabla_{\boldsymbol{\theta}} f_{\boldsymbol{\theta}}(\mathbf{x}, \hat{y})$$
$$= \mathbb{E}_{p_d(\mathbf{x}, y)} \left[ \nabla_{\boldsymbol{\theta}} f_{\boldsymbol{\theta}}(\mathbf{x}, y) - \mathbb{E}_{p_{\boldsymbol{\theta}}(\hat{y}|\mathbf{x})} \nabla_{\boldsymbol{\theta}} f_{\boldsymbol{\theta}}(\mathbf{x}, \hat{y}) \right]$$
$$= \mathbb{E}_{p_d(\mathbf{x}, y)} \left[ \nabla_{\boldsymbol{\theta}} f_{\boldsymbol{\theta}}(\mathbf{x}, y) - \sum_{k=1}^{K} p_{\boldsymbol{\theta}}(k|\mathbf{x}) \nabla_{\boldsymbol{\theta}} f_{\boldsymbol{\theta}}(\mathbf{x}, k) \right]$$
$$= \mathbb{E}_{p_d(\mathbf{x}, y)} \left[ \nabla_{\boldsymbol{\theta}} f_{\boldsymbol{\theta}}(\mathbf{x}, y) - \sum_{k=1}^{K} \frac{\exp(f_{\boldsymbol{\theta}}(\mathbf{x}, k)) \nabla_{\boldsymbol{\theta}} f_{\boldsymbol{\theta}}(\mathbf{x}, k)}{\sum_{j=1}^{K} \exp(f_{\boldsymbol{\theta}}(\mathbf{x}, j))} \right]$$
$$= \mathbb{E}_{p_d(\mathbf{x}, y)} \left[ \nabla_{\boldsymbol{\theta}} f_{\boldsymbol{\theta}}(\mathbf{x}, y) - \sum_{k=1}^{K} \frac{\nabla_{\boldsymbol{\theta}} \exp(f_{\boldsymbol{\theta}}(\mathbf{x}, k))}{\sum_{j=1}^{K} \exp(f_{\boldsymbol{\theta}}(\mathbf{x}, j))} \right]$$
$$= \mathbb{E}_{p_d(\mathbf{x}, y)} \left[ \nabla_{\boldsymbol{\theta}} f_{\boldsymbol{\theta}}(\mathbf{x}, y) - \frac{\nabla_{\boldsymbol{\theta}} \sum_{k=1}^{K} \exp(f_{\boldsymbol{\theta}}(\mathbf{x}, k))}{\sum_{j=1}^{K} \exp(f_{\boldsymbol{\theta}}(\mathbf{x}, j))} \right]$$
$$= \mathbb{E}_{p_d(\mathbf{x}, y)} \left[ \nabla_{\boldsymbol{\theta}} f_{\boldsymbol{\theta}}(\mathbf{x}, y) - \nabla_{\boldsymbol{\theta}} \log \sum_{k=1}^{K} \exp(f_{\boldsymbol{\theta}}(\mathbf{x}, k)) \right]$$
$$= \mathbb{E}_{p_d(\mathbf{x}, y)} \nabla_{\boldsymbol{\theta}} \log \frac{\exp(f_{\boldsymbol{\theta}}(\mathbf{x}, y))}{\sum_{k=1}^{K} \exp(f_{\boldsymbol{\theta}}(\mathbf{x}, k)}$$
$$= \nabla_{\boldsymbol{\theta}} \mathbb{E}_{p_d(\mathbf{x}, y)} \log p_{\boldsymbol{\theta}}(y|\mathbf{x}). \tag{17}$$

## C.4. Equivalence between InfoNCE Loss and Non-parametric CEM

In Section A.1, we have claimed that the the log likelihood gradient of NP-CEM equals to exactly the negative gradient of the InfoNCE loss when we approximate $p_{\boldsymbol{\theta}}(\mathbf{x})$ with $p_d(\mathbf{x})$. The derivation is presented as follows:

$$\mathbb{E}_{p_d(\mathbf{x},\mathbf{z})}\nabla_{\boldsymbol{\theta}}f_{\boldsymbol{\theta}}(\mathbf{x},\mathbf{z}) - \mathbb{E}_{p_{\boldsymbol{\theta}}(\hat{\mathbf{x}},\hat{\mathbf{z}})}\nabla_{\boldsymbol{\theta}}f_{\boldsymbol{\theta}}\left(\hat{\mathbf{x}},\hat{\mathbf{z}}\right)$$

$$=\mathbb{E}_{p_d(\mathbf{x},\mathbf{z})}\nabla_{\boldsymbol{\theta}}f_{\boldsymbol{\theta}}(\mathbf{x},\mathbf{z}) - \mathbb{E}_{p_{\boldsymbol{\theta}}(\mathbf{x})p_{\boldsymbol{\theta}}(\mathbf{z}|\mathbf{x})}\nabla_{\boldsymbol{\theta}}f_{\boldsymbol{\theta}}\left(\hat{\mathbf{x}},\hat{\mathbf{z}}\right)$$

$$=\mathbb{E}_{p_d(\mathbf{x},\mathbf{z})}\nabla_{\boldsymbol{\theta}}f_{\boldsymbol{\theta}}(\mathbf{x},\mathbf{z}) - \mathbb{E}_{p_{\boldsymbol{\theta}}(\hat{\mathbf{x}})p_{\boldsymbol{\theta}}(\hat{\mathbf{z}}|\hat{\mathbf{x}})}\nabla_{\boldsymbol{\theta}}f_{\boldsymbol{\theta}}\left(\hat{\mathbf{x}},\hat{\mathbf{z}}\right)$$

$$=\mathbb{E}_{p_d(\mathbf{x},\mathbf{z})}\nabla_{\boldsymbol{\theta}}f_{\boldsymbol{\theta}}(\mathbf{x},\mathbf{z}) - \mathbb{E}_{p_{\boldsymbol{\theta}}(\hat{\mathbf{x}})p_{\boldsymbol{\theta}}(\hat{\mathbf{z}})}\frac{p_{\boldsymbol{\theta}}(\hat{\mathbf{z}}|\hat{\mathbf{x}})}{p_{\boldsymbol{\theta}}(\hat{\mathbf{z}})}\nabla_{\boldsymbol{\theta}}f_{\boldsymbol{\theta}}(\hat{\mathbf{x}},\hat{\mathbf{z}})$$

$$\approx\mathbb{E}_{p_d(\mathbf{x},\mathbf{z})}\nabla_{\boldsymbol{\theta}}f_{\boldsymbol{\theta}}(\mathbf{x},\mathbf{z}) - \mathbb{E}_{p_d(\hat{\mathbf{x}})p_d(\hat{\mathbf{z}})}\frac{p_{\boldsymbol{\theta}}(\hat{\mathbf{z}}|\hat{\mathbf{x}})}{p_{\boldsymbol{\theta}}(\hat{\mathbf{z}})}\nabla_{\boldsymbol{\theta}}f_{\boldsymbol{\theta}}(\hat{\mathbf{x}},\hat{\mathbf{z}})$$

$$=\mathbb{E}_{p_d(\mathbf{x},\mathbf{z})}\left[\nabla_{\boldsymbol{\theta}}f_{\boldsymbol{\theta}}(\mathbf{x},\mathbf{z}) - \mathbb{E}_{p_d(\hat{\mathbf{z}})}\frac{p_{\boldsymbol{\theta}}(\hat{\mathbf{z}}|\hat{\mathbf{x}})}{p_{\boldsymbol{\theta}}(\hat{\mathbf{z}})}\nabla_{\boldsymbol{\theta}}f_{\boldsymbol{\theta}}(\mathbf{x},\mathbf{z})\right]$$

$$\approx\frac{1}{N}\sum_{i=1}^{N}\left[\nabla_{\boldsymbol{\theta}}f_{\boldsymbol{\theta}}(\mathbf{x}_i,\mathbf{z}_i) - \sum_{k=1}^{K}\frac{\exp(f_{\boldsymbol{\theta}}(\mathbf{x},\mathbf{z}_{ik}^-))\nabla_{\boldsymbol{\theta}}f_{\boldsymbol{\theta}}\left(\mathbf{x},\mathbf{z}_{ik}^-\right)}{\sum_{k}\exp(f_{\boldsymbol{\theta}}(\mathbf{x},\mathbf{z}_{ik}^-))}\right]$$

$$=\frac{1}{N}\sum_{i=1}^{N}\left[\nabla_{\boldsymbol{\theta}}f_{\boldsymbol{\theta}}(\mathbf{x}_i,\mathbf{z}_i) - \nabla_{\boldsymbol{\theta}}\log\sum_{k=1}^{K}\exp(f_{\boldsymbol{\theta}}(\mathbf{x},\mathbf{z}_{ik}^-))\right]$$

$$\approx\frac{1}{N}\sum_{i=1}^{N}\nabla_{\boldsymbol{\theta}}\log\frac{\exp(f_{\boldsymbol{\theta}}(\mathbf{x}_i,\mathbf{z}_i))}{\sum_{k=1}^{K}\exp(f_{\boldsymbol{\theta}}(\mathbf{x}_i,\mathbf{z}_{ik}^-))}. \tag{18}$$

## C.5. Equivalence between Adversarial InfoNCE and NP-CEM

In Section A.2, we have developed the unsupervised analogy of AT loss and regularization. In particular, we have claimed that contrastive gradient is equivalent to the gradient of the Adversarial InfoNCE loss (i.e., the InfoNCE loss of the adversarial example $\hat{\mathbf{x}}$) utilized in previous work (Jiang et al., 2020; Ho & Nvasconcelos, 2020; Kim et al., 2020). It can be derived following Eqn. (18):

$$\mathbb{E}_{p_d(\mathbf{x},\mathbf{z})\otimes p_{\boldsymbol{\theta}}(\hat{\mathbf{x}},\hat{\mathbf{z}})}\left[\nabla_{\boldsymbol{\theta}}f_{\boldsymbol{\theta}}(\hat{\mathbf{x}},\mathbf{z}) - \nabla_{\boldsymbol{\theta}}f_{\boldsymbol{\theta}}(\hat{\mathbf{x}},\hat{\mathbf{z}})\right]$$

$$=\mathbb{E}_{p_d(\mathbf{x},\mathbf{z})\otimes p_{\boldsymbol{\theta}}(\hat{\mathbf{x}})}\left[\nabla_{\boldsymbol{\theta}}f_{\boldsymbol{\theta}}(\hat{\mathbf{x}},\mathbf{z}) - \mathbb{E}_{p_{\boldsymbol{\theta}}(\hat{\mathbf{z}}|\hat{\mathbf{x}})}\nabla_{\boldsymbol{\theta}}f_{\boldsymbol{\theta}}(\hat{\mathbf{x}},\hat{\mathbf{z}})\right]$$

$$\approx\mathbb{E}_{p_d(\mathbf{x},\mathbf{z})\otimes p_{\boldsymbol{\theta}}(\hat{\mathbf{x}})}\left[\nabla_{\boldsymbol{\theta}}f_{\boldsymbol{\theta}}(\hat{\mathbf{x}},\mathbf{z}) - \mathbb{E}_{p_d(\hat{\mathbf{z}})}\frac{p_{\boldsymbol{\theta}}(\hat{\mathbf{z}}|\hat{\mathbf{x}})}{p_{\boldsymbol{\theta}}(\hat{\mathbf{z}})}\nabla_{\boldsymbol{\theta}}f_{\boldsymbol{\theta}}(\hat{\mathbf{x}},\hat{\mathbf{z}})\right]$$

$$\approx\frac{1}{N}\sum_{i=1}^{N}\nabla_{\boldsymbol{\theta}}\log\frac{\exp(f_{\boldsymbol{\theta}}(\hat{\mathbf{x}}_i,\mathbf{z}_i))}{\sum_{k=1}^{K}\exp(f_{\boldsymbol{\theta}}(\hat{\mathbf{x}}_i,\mathbf{z}_{ik}^-))}. \tag{19}$$