# OpenReview forum: "Demystifying Adversarial Training via A Unified Probabilistic Framework"
_ICML.cc/2021/Workshop/AML — ICML 2021 Workshop AML Poster_

### Official Review · Reviewer_aoAm · 2021-06-20
**Interesting work**

**Rating:** Accept
**Confidence:** 5

**Review:**

It is a promising topic to explore the connection between AT and generative models (e.g., EBMs), based on the observations like semantic gradients in adversarially trained models. The experiments are also comprehensive, involving several recent works on score-based models. Related topics are worthy of more attention by the adversarial community.

---

### Decision · Program_Chairs · 2021-06-21

**Decision:**

Accept (Poster)

**Comment:**

This paper studied a promising topic to explore AT and generative models. The experiments are comprehensive.